# Epigenetics and chromatin structure regulate *var2csa* expression and the placental-binding phenotype in *Plasmodium falciparum*

Todd Lenz[1†], Madle Sirel[2†], Hannes Hoppe[2], Sulman Shafeeq[2], Karine G Le Roch[1*†], Ulf Ribacke[2,3†]

[1]Department of Molecular, Cell and Systems Biology, University of California, Riverside, Riverside, United States; [2]Department of Microbiology, Tumor and Cell Biology (MTC), Karolinska Institutet, Solna, Sweden; [3]Department of Cell and Molecular Biology (ICM), Uppsala University, Uppsala, Sweden

## eLife Assessment

This interesting study presents a multi-OMICs approach to unify different lines of evidence regarding the epigenetic regulation of the key virulence factor causing placental malaria during *P. falciparum* infection. Most results are confirmatory of previous observations; nonetheless, the claims are supported by **convincing** evidence. The combinatorial approach chosen here is unprecedented and therefore provides **valuable** new data. In addition, the comparative investigation of different DNA methylation modifications is novel and disproves a direct role in *var* gene regulation.

**Abstract** *Plasmodium falciparum* is responsible for what appears to be a never-ending public health issue in the developing world. With repeated infections, a gradual semi-immunity to severe malaria can be acquired, but this is disrupted when women become pregnant as the parasite cytoadheres in the placenta to prevent splenic clearance. This change in tissue tropism is due to specific transcription of the antigenically variable adhesin VAR2CSA. To better understand the molecular mechanisms activating *var2csa* and antigenic variation overall, we used a combination of phenotypic and systems biology assays. We first established phenotypically homogenous populations of VAR2CSA-expressing and placenta-binding parasites that were shown to exclusively transcribe *var2csa* while all other *var* genes remained silenced. We also confirmed that the transcriptional activation was strongly associated with distinct depletion of repressive H3K9me3 marks. Further, we used chromatin conformation capture as a high-resolution approach to determine interchromosomal interactions and established that transcriptional activation is linked to a small yet significant repositioning of *var2csa* relative to heterochromatic telomeric clusters. Lastly, we demonstrated that occupancy of 5-methylcytosine was present in all *var* genes but independent of transcriptional repression and switching. All together, these findings provide insights at high resolution into the potential role of 5-methylcytosine in *P. falciparum* and increase our understanding of the mechanisms regulating antigenic variation at the epigenetics and chromatin structure level.

## Introduction

The morbidity and mortality associated with *Plasmodium falciparum* malaria are negatively correlated to the number of infections contracted (*Doolan et al., 2009*). As a consequence, young children and

**\*For correspondence:**
karine.leroch@ucr.edu

[†]These authors contributed equally to this work

**Competing interest:** The authors declare that no competing interests exist.

other malaria-naïve individuals are at greatest risk of developing severe illness and succumbing to the disease. With repeated exposure, a gradual immunity to clinical manifestations is acquired and symptomatic episodes in adults are relatively rare in endemic areas (*Doolan et al., 2009*). This holds true until women become pregnant, when the built-up immunity is disrupted and they again become highly susceptible to developing severe disease (*Steketee et al., 2001*). Besides affecting the mother, placental malaria (PM) also results in adverse outcomes for the fetus, including miscarriages, stillbirths, preterm births, and low birth weights, of which the latter two often result in predisposition for morbidities later in life (*Rogerson and Beeson, 1999*; *Nyirjesy et al., 1993*; *Chua et al., 2021*). Thus, PM represents a devastating public health problem that targets the most vulnerable populations in the resource-scarce malaria-endemic regions.

The onset of PM is tightly linked to the appearance of the placenta, which presents a new niche for the parasitized red blood cell (pRBC) to sequester and thrive. To cytoadhere in the microvasculature, the parasite employs antigenically variable adhesins from the *var* gene encoded *P. falciparum* Erythrocyte Membrane Protein 1 (PfEMP1) family of proteins, a key virulence feature of the parasite that is directly linked to the pathogenesis of severe disease (*Scherf et al., 2008*; *Wahlgren et al., 2017*). While harmful for the human host, the cytoadhesion provides the parasite immune evasive opportunities, such as sequestration of pRBC to deep vascular walls to avoid splenic clearance or resetting, the adhesion of pRBCs to uninfected RBCs to mask antigens and prevent immune cells from recognizing the pRBCs. (*Lee et al., 2019*). The repertoire of PfEMP1 is large, with approximately 60 *var* genes per parasite genome from which only one is assumed to be expressed at a time, a process referred to as mutually exclusive expression. Transcriptional switching between the different *var* loci creates antigenic variation, which leads to immune evasion and binding of a plethora of human receptors and variable tissue tropism for the parasite (*Lee et al., 2019*; *Chen et al., 1998*; *Scherf et al., 1998*). In between parasites, the genetic variation among the absolute majority of *var* genes is vast, with one important exception, the remarkably well-conserved *var2csa*. This gene encodes the PfEMP1 VAR2CSA that mediates cytoadhesion to chondroitin sulfate A (CSA) on syncytiotrophoblasts in the placenta and is considered the main culprit behind PM (*Fried and Duffy, 1996*; *Salanti et al., 2004*).

The variable expression of PfEMP1 is thought to be accomplished through several layers of gene regulatory activities to ensure that only one, or very few, *var* genes are translated to functional adhesins and exported to the surface of the pRBC (*Chen et al., 1998*; *Scherf et al., 1998*; *Brolin et al., 2009*; *Joergensen et al., 2010*; *Hollin and Le Roch, 2020*). Recent bulk transcriptomics and single-cell RNA-seq have demonstrated that var gene switching doesn't seem purely random; instead, parasite populations tend to progress through successive waves of parasitemia, with each wave dominated by parasites expressing one (or a few) var genes before the system resolves back to one predominant var gene (16 and 17). The antigenic variation is mainly achieved by transcriptional regulation, where altered DNA accessibility is orchestrated through variations in the nucleosome composition, histone modifications, and organization of the chromatin into active and repressive clusters. In addition, involvement of non-coding RNAs and specific transcription factors has been proposed to compose additional layers in the regulation that governs repression, activation, and switching of transcription into one interconnected regulatory network, enabling coordinated expression patterns across many infected red blood cells (*Hollin and Le Roch, 2020*; *Zhang et al., 2022*; *Florini et al., 2025*). Lastly, the peculiar VAR2CSA appears also regulated on a translational level (*Mok et al., 2008*; *Amulic et al., 2009*; *Chan et al., 2017*).

While major advances have been made recently to better understand gene transcription of these important virulence factors, mechanisms regulating these events at the molecular level are complex and many questions remain. An improved understanding of the detailed molecular mechanisms that mediate parasite adhesion is vital if we want to grasp the molecular factors that control malaria pathology and to identify new therapeutic strategies. Here, using a combination of advanced phenotypic assays and systems biology approaches including transcriptomics, epigenetics, and chromatin structure features, we provide a detailed view in high-resolution on how expression of the PM mediating *var2csa* is regulated at transcriptional initiation level. Intriguingly, we also show that occupancy of the epigenetic mark 5-methylcytosine (5mC) is not only found in most gene bodies but is also present at higher levels in highly expressed genes, including antigenically variable genes. However, the 5mC mark appears to be unrelated to transcriptional activation and switching.

## Results

### Repeated in vitro selection on CSA results in a highly homogenous population of VAR2CSA-expressing and placenta-binding parasites

*P. falciparum* pRBCs have different receptor preferences for cytoadhesion to diverse host cells. This has been previously established in vitro by panning on antibody-coated beads or either cell-bound or soluble human receptors (*Brolin et al., 2009*; *Roberts et al., 1992*; *Staalsoe et al., 2003*). While several parasite molecules have been identified as ligands for cytoadhesion, the antigenically variant PfEMP1 proteins, encoded by the *var* genes, have been demonstrated as the genuine adhesion molecules. In PM, VAR2CSA encoded by a subfamily of the *var* genes plays a vital role for the cytoadhesion of pRBCs to the CSA expressed on the surface of placental syncytiotrophoblasts. To better understand the molecular mechanisms regulating *var2csa* transcription, we first employed an approach that entailed several rounds of repeated panning of NF54 pRBCs on chondroitin sulfate A (CSA)-coated plates (*Brolin et al., 2009*) to select for parasites with homogenous expression of VAR2CSA and a placenta-binding phenotype. Prior to the panning procedure, we confirmed by scanning electron microscopy (SEM) the presence of knob structures, which are protrusions where PfEMP1s are surface exposed on pRBCs in patient isolates but are easily lost upon long-term in vitro cultivation (*Figure 1A*). Several phenotypic analyses were used to confirm the accuracy of our selection process. We analyzed pRBC surface expression of VAR2CSA by flow cytometry-based antibody recognition throughout the trajectory of repeated panning and revealed a gradual transition from a phenotypically heterogeneous to homogeneous parasite population (*Figure 1B*). In addition, we interrogated the cytoadhesive potential using placental sections from malaria non-immune donors (*Flick et al., 2001*). For the repeatedly panned parasites, substantial numbers of pRBCs were observed binding to the placental syncytiotrophoblasts (*Figure 1C*). We were able to quantify bound pRBCs per mm2 using a VAR2CSA surface negative PTEF knockout parasite (*Chan et al., 2017*) as control for background binding and revealed an approximate sixfold higher binding of the CSA-selected parasite line (*Figure 1D*). Thus, the repeatedly panned NF54CSA (from here on referred to as NF54CSAh) was deemed phenotypically highly homogenous with a VAR2CSA/PM relevant cytoadhesion profile and considered suitable for downstream analyses of *var2csa* transcription and regulation.

### The *P. falciparum* placental-binding phenotype is strictly linked to transcription of *var2csa*

Next, we sought to decipher any transcriptomic differences responsible for the phenotypic disparity observed between the unselected NF54 and NF54CSAh. As PfEMP1 encoding *var* genes are actively transcribed during the first third of the parasite's 48 h life cycle (also referred to as the ring stage) (*Bozdech et al., 2003*; *Le Roch et al., 2003*), we harvested NF54 and NF54CSAh from three independent selections at 16±2 h post invasion (p.i.) of RBCs for RNA-seq. We also collected parallel samples for ChIP-seq, MeDIP-seq, and Hi-C.

The RNA-seq analysis revealed differential transcription in only 64 of the 5285 protein coding genes of the parasite (*Supplementary file 1*). Hierarchical clustering and principal component analysis performed on the complete set of data revealed little deviation ($\rho > 0.99$) between both replicates and samples when accounting for genome-wide transcription (*Figure 1—figure supplement 1A and B*). We, however, observed a very strong correlation between replicates of NF54 ($\rho > 0.97$) and between replicates of NF54CSAh ($\rho > 0.63$) when only taking *var* genes into account (*Figure 1E*), confirming a high degree of phenotypic maintenance among collected samples. In line with this, we observed a near-exclusive representation of *var* genes among differentially expressed genes (*Figure 1F*), with *var2csa* being the only upregulated and almost all other of the 60 *var* genes downregulated in NF54CSAh compared to NF54 (*Figure 1F and G*). Besides the confirmation of *var2csa* transcription being closely associated to the placental-binding phenotype of malaria parasites, the high degree of phenotypic homogeneity of NF54CSAh and the extensive heterogeneity of NF54 provided an excellent opportunity for downstream analyses of gene regulation in antigenic variation via epigenetic mechanisms and chromatin structure.

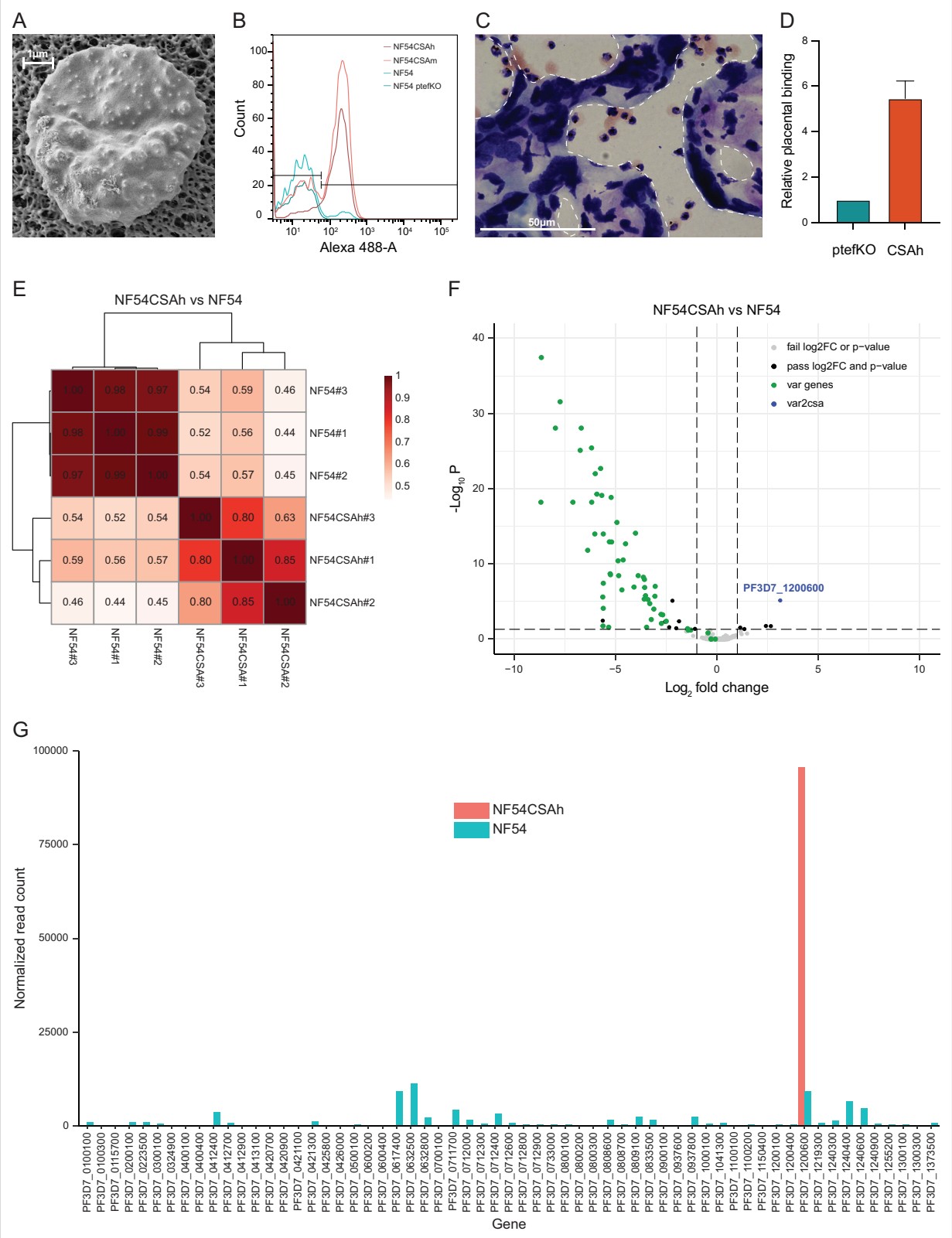

**Figure 1.** Generation of parasites with a homogenous placental-binding phenotype and consequential dominant expression of *var2csa*. (**A**) The presence of in vivo relevant knob structures on pRBCs was confirmed by SEM prior to phenotypic selection. Shown is a representative image of a pRBC with knobs appearing as white protrusions. (**B**) Flow cytometry-based antibody recognition of VAR2CSA surface expression on repeatedly selected pRBCs. NF54 ptefKO served as negative control, NF54 was the original and unselected parasite line, NF54CSAm was an intermediately selected line

*Figure 1 continued on next page*

*Figure 1 continued*

and NF54CSAh was panned enough to achieve a homogenous population of VAR2CSA-expressing pRBCs. (**C**) Representative image of a Giemsa-stained placental section with bound NF54CSAh pRBCs. Tissue boundaries are indicated by white dashed lines. (**D**) Quantification of relative placental binding of NF54CSAh from three biological replicates. pRBCs bound per mm$^2$ were normalized to the VAR2CSA negative NF54 ptefKO parasite line (n=3). (**E**) Spearman correlation of *var* gene expression profiles determined by RNA-seq for three bio-replicates each of NF54CSAh and the original, unselected NF54. (**F**) Differential gene expression between NF54CSAh and NF54 identified a limited number of significant genes (FDR <0.05, Log$_2$ fold change >1) of which the majority were *var* genes (*var2csa* in blue, all other *var* genes in green). (**G**) Normalized read counts for all *var* genes revealed NF54 to be phenotypically highly heterogeneous, whereas *var2csa* was the only *var* gene expressed by NF54CSAh.

The online version of this article includes the following figure supplement(s) for figure 1:

**Figure supplement 1.** Genome-wide expression profiles are highly similar among all samples.

## Transcriptional activity of *var2csa* is associated with a near-complete lack of H3K9me3 occupancy

Besides its known role in repressing repetitive elements and gene-depleted regions in eukaryotes, the histone H3 lysine 9 trimethylation mark (H3K9me3) has been demonstrated to be an important player in heterochromatin formation, to silence lineage-inappropriate genes and control cell fate. This appears particularly true for *Plasmodium* spp., where H3K9me3 marks have been identified as potentially critical in regulating parasite-specific genes involved in pathogenicity and sexual commitment (*Ninova et al., 2019*; *Lopez-Rubio et al., 2009*). We therefore utilized chromatin ChIP-seq as previously described (*Bunnik et al., 2018*) to assess the distribution of H3K9me3 and examine whether NF54CSAh displays differentially bound heterochromatin/euchromatin marks due to elevated transcription of *var2csa*. We performed ChIP-seq experiments in triplicates using NF54CSAh and NF54 samples collected at the same time as the samples used for the RNA-seq experiments described above. ChIP-seq libraries were sequenced, processed, and mapped to the genome. Following input normalization, we observed a clear targeted enrichment of H3K9me3 within both subtelomeric and internal *var* gene clusters as previously described (*Michel-Todó et al., 2023*; *Figure 2A*). H3K9me3 occupancy within the gene body of *var2csa* of NF54CSAh was almost entirely eliminated, differentiating it from the NF54 control which shows little variation from other *var* genes, which goes in line with its highly heterogeneous *var* phenotype and consequently small contribution to H3K9me3 for individual genes on a population level (*Figure 2A and B*). Due to highly homologous sequences preventing unique mapping, several *var* genes display no H3K9me3 coverage (*Figure 2B*).

This observation was confirmed by performing MACS3 broad peak calling of input normalized H3K9me3 enrichment with results showing an average of 2234 significant consensus peaks (q<0.05) per sample and approximately 79% of peaks within the coding sequence (CDS) of protein coding genes (*Supplementary file 2*). Differential binding analysis via DiffBind showed strong correlation (*r*>0.75) between replicates and samples (*Figure 2—figure supplement 1A and B*), with only four differentially bound sites (FDR <0.05) (*Figure 2—figure supplement 1C, D and E*), all within the CDS or just upstream of the transcription start site (TSS) of *var2csa* (*Supplementary file 2*). Given that *var2csa* is significantly downregulated in NF54, these results demonstrate the correlation between loss of *var2csa* expression and H3K9me3-mediated transcriptional silencing.

*P. falciparum* variant antigen gene families such as *var, rif, stevor,* and *pfmc-2tm* cluster within subtelomeric regions of most chromosomes and enrichment of H3K9me3 within the gene body is similarly linked to their repression (*Lopez-Rubio et al., 2007*; *Salcedo-Amaya et al., 2009*; *Howitt et al., 2009*). NF54CSAh and NF54 both display this same pattern of elevated H3K9me3 within the exons of these multicopy gene families, and after normalizing each region (5′ UTR, exons, introns, and intergenic) the mean H3K9me3 within each is higher than genes outside these families as well as all intergenic regions (*Figure 2C and D*).

## Increased interchromosomal interaction and perinuclear repositioning are involved in silencing of *var* genes and activation of *var2csa*, respectively

To further investigate the effect chromatin organization and accessibility has on transcription of highly variant multi-copy gene families such as *var* and *rif*, we performed Hi-C experiments on tightly synchronized trophozoites to ensure peak expression of the target genes. Hi-C libraries of three biological

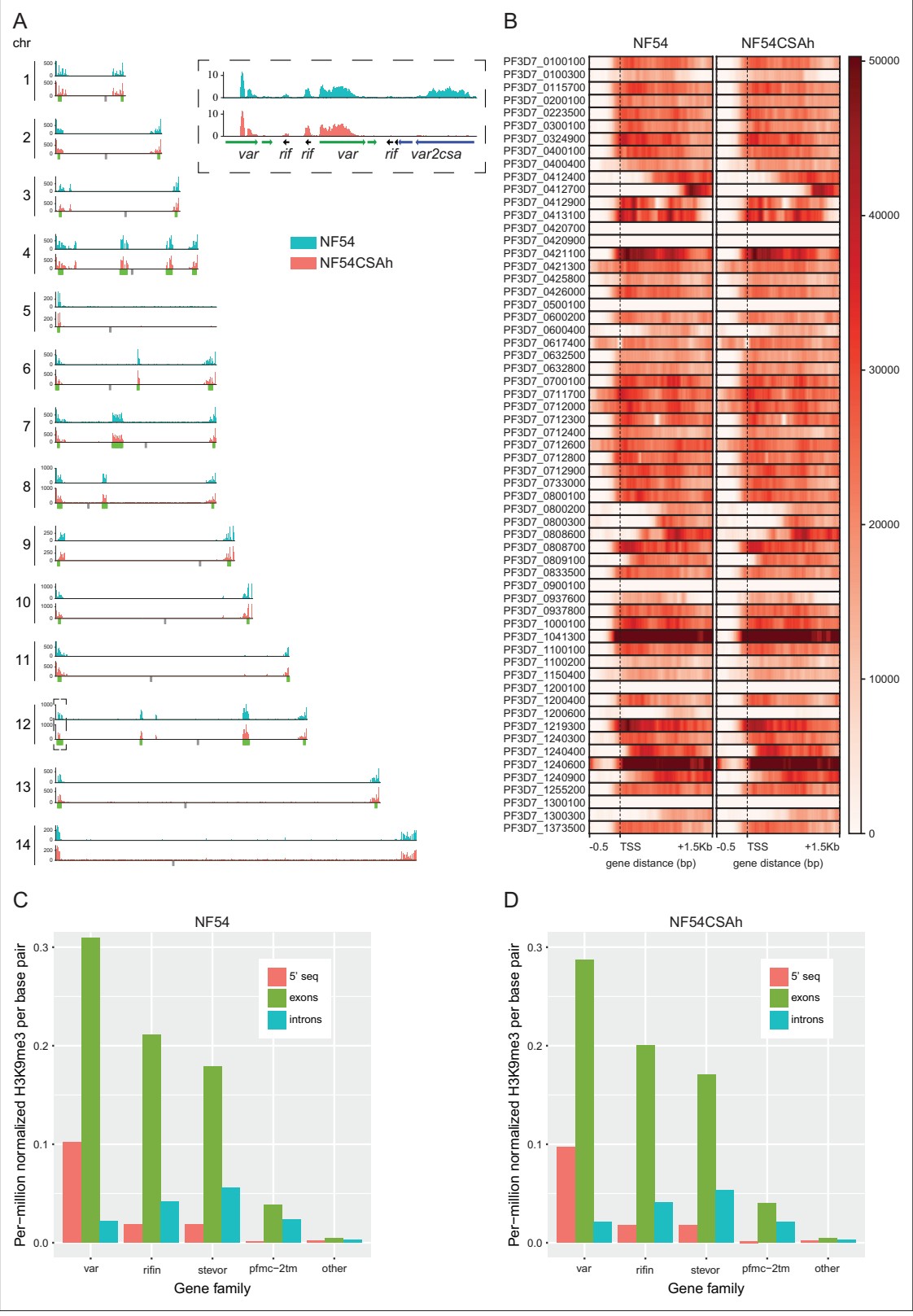

**Figure 2.** The *var2csa* locus is devoid of repressive H3K9me3 in placental-binding parasites. (**A**) Chromosomal distribution of repressive H3K9me3 as determined by ChIP-seq from merged biological triplicates of NF54 and NF54CSAh. Regional clusters of antigenically variable genes are noted with green below each plot. The only noticeable difference between the parasite lines was located to the left arm of chromosome 12, which contains the *var2csa* locus (dashed box). (**B**) Heatmaps of H3K9me3 occupancy in NF54 and NF54CSAh for all *var* genes (0.5 kb upstream of the transcription start site

*Figure 2 continued on next page*

*Figure 2 continued*

(TSS) and 1.5 kb into the gene body) with *var2csa* displayed with bolded gene ID. (**C, D**) H3K9me3 enrichment in the antigenically variable multigene families *var*, *rif*, *stevor*, and *pfmc-2tm* partitioned into 5' UTR, exons, and introns versus the rest of the genomes (all other genes and intergenic regions) of NF54 and NF54CSAh.

The online version of this article includes the following figure supplement(s) for figure 2:

**Figure supplement 1.** Differential peak calling of H3K9me3 reveals highly focused changes in *var2csa* region.

replicates for each sample (NF54 and NF54CSAh) were prepared as previously described (***Bunnik et al., 2018***; ***Gupta et al., 2021***) and sequenced to a mean depth of ~155 million reads per replicate. The libraries were processed (aligning, pairing, mapping, and quality filtering) using HiC-Pro (***Servant et al., 2015***) and resulted in a mean of ~44 million valid interaction pairs per replicate. Due to the *P. falciparum* genome size and frequency of MboI cut sites, we elected to bin our reads at a 10 kb resolution to identify intrachromosomal and interchromosomal interactions.

A high stratum-adjusted correlation coefficient (SCC ≈ 0.87–0.94) suggests that chromatin structures were highly similar between replicates and samples (***Figure 3—figure supplement 1***). We therefore combined the biological replicates for downstream analyses. Due to variation in sequencing depth between the merged NF54 and NF54CSAh samples, random sampling was performed on NF54CSAh to obtain ~100 million consensus reads for comparative and differential analyses. Heatmaps generated from the ICED normalized matrices show patterns similar to previous studies (***Ay et al., 2014***; ***Bunnik et al., 2018***; ***Bunnik et al., 2019***), with a negative log-linear relationship between contact probability and genomic distance demonstrating that our experiment worked as expected (***Figure 3—figure supplement 1***). Most intrachromosomal interactions occur at a distance less than 10% the total length of each chromosome, with heterochromatin clustering occurring in telomeric regions and internal *var* gene clusters at a higher frequency than the other distant regions (***Figure 3A–C***, ***Figure 3—figure supplements 2 and 3***).

To evaluate the correlation between transcription and chromatin architecture, we used Selfish (***Ardakany et al., 2019***) to identify differential intrachromosomal and interchromosomal interactions. Although there were slight variations in the number of intrachromosomal interactions across most regions, the consistent pattern that emerged was the increased number of interactions between subtelomeric regions on most chromosomes ($p<0.05$, $\log_2FC >1$) in NF54CSAh over NF54, indicating tighter heterochromatin control of *var* gene regions (***Figure 3A–C*** and ***Figure 3—figure supplement 4***). The subtelomeric regions on chromosomes 2, 3, 4, and 10 showed the largest increase in interaction frequency, and all the *var* genes within those regions were significantly down-regulated in the RNA-seq data generated in this study. Most interesting is perhaps the mix of increased and decreased interactions between subtelomeric regions on chromosome 12, due to the proximity of *var2csa* to other nearby *var* genes within that region confirming a potential role of the chromatin 3D structure in gene regulation.

Additional analysis of these small-scale structural changes was performed by 3D chromatin modeling using the Poisson-based algorithm, PASTIS (***Varoquaux et al., 2014***). Modeling shows co-localization of centromeres and telomeric clustering in distinct regions within the nucleus (***Figure 3D and E***). The overall 3D structure of the genome was similar between NF54 and NF54CSAh. Using the coordinates output from PASTIS, we computed the distance between various bins/regions and found that the mean distance between telomeres containing *var* genes decreased by 6% in NF54CSAh, indicating overall telomeric compaction. What is more significant is that although there is a 52% increase in distance of *var2csa* to the nearest bin in spatial proximity and 19% greater distance from the telomeric cluster, the *var* gene nearest to the end of chromosome 12 only shows a 3% increase in distance from the telomeric cluster. This stark difference in distances for two regions only separated by 20 kb shows that small and localized changes in the chromatin structure are enough to allow transcriptional activation of *var2csa*.

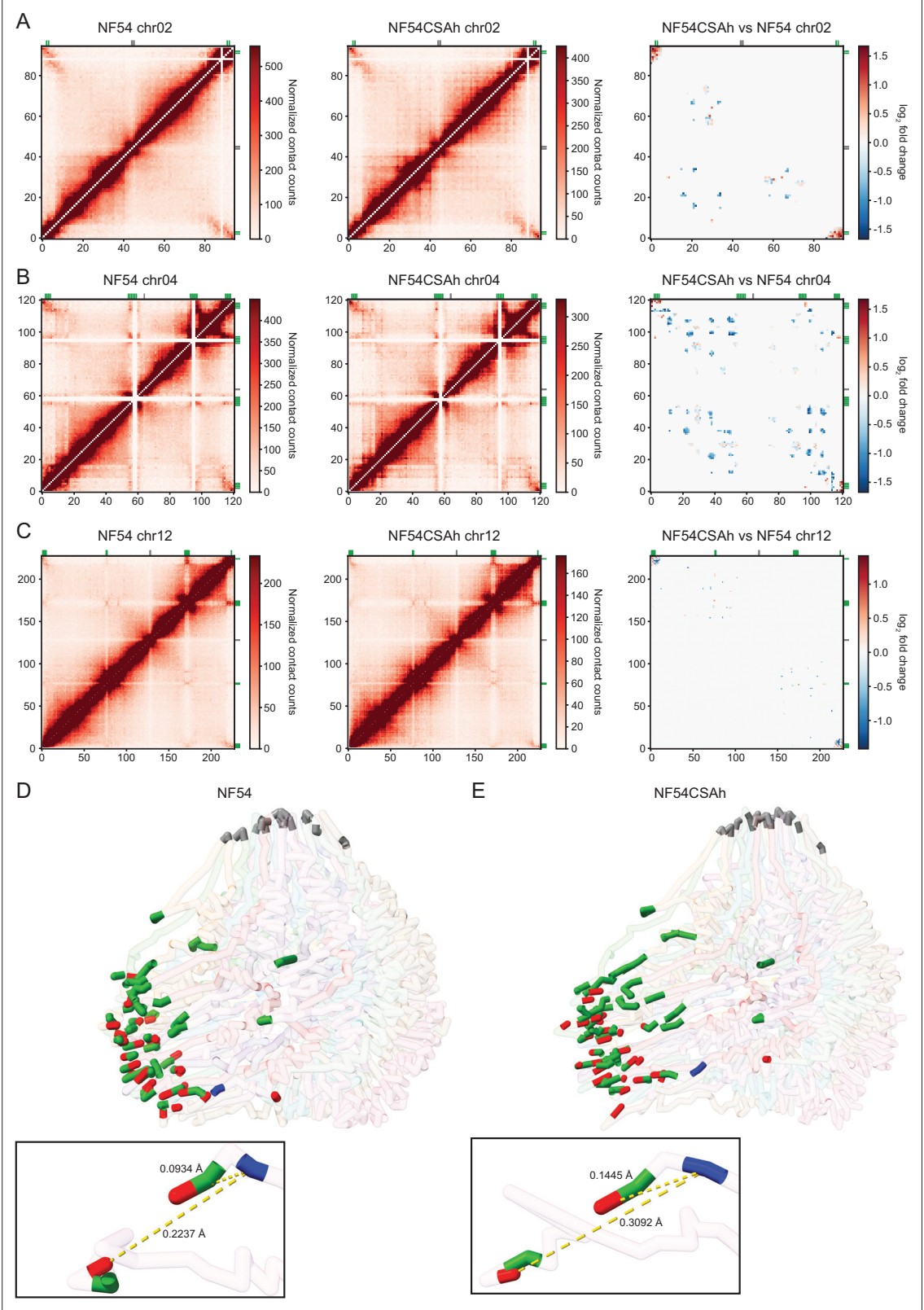

**Figure 3.** Changes in chromatin organization and perinuclear repositioning upon var2csa activation and silencing. (**A–C**) Hi-C generated and normalized contact counts representing intrachromosomal interactions for chromosomes 2, 4, and 12 in NF54 (left panels) and NF54CSAh (middle panels). Differential intrachromosomal interactions between the two parasite lines (right panels) displayed a general increase in interactions between subtelomeric and chromosome internal *var* gene clusters for NF54CSAh compared to NF54. Chromosome 12 (**C**) stood out as the only exception with

*Figure 3 continued on next page*

*Figure 3 continued*

decreased interactions in NF54CSAh for the subtelomere containing the *var2csa* locus. (**D**) 3D chromatin modeling for NF54 displays a polarized nucleus with clustering of centromeres (gray) and telomeres (red) in distinct regions. The majority of *var* genes (green), including *var2csa* (blue), are located in close vicinity to the telomeric cluster. (**E**) The 3D chromatin model for NF54CSAh displayed a similar topology to NF54 with the exception of decreased distance between *var* genes (green) and telomere ends (red) overall and an increased distance of *var2csa* (blue) from the telomeric cluster.

The online version of this article includes the following figure supplement(s) for figure 3:

**Figure supplement 1.** There is a strong correlation in genome-wide interactions between samples and replicates and between contact count probability and genomic distance.

**Figure supplement 2.** Hi-C intrachromosomal and interchromosomal contact count heatmaps for NF54.

**Figure supplement 3.** Hi-C intrachromosomal and interchromosomal contact count heatmaps for NF54CSAh.

**Figure supplement 4.** Differential interaction contact count heatmaps show higher interactions between telomeric and internal *var* gene-containing regions in NF54CSAh.

## Distribution of DNA methylation may influence gene expression overall but does not mediate transcriptional activation and switching in antigenic variation

DNA methylation of cytosine residues serves yet another epigenetic transcriptional antagonist and is found in many model eukaryotic organisms; however, the context and level of methylation differs among plants and animals (*Zhang et al., 2006*; *Lister et al., 2009*; *Feng et al., 2010*; *Zemach et al., 2010*). In higher eukaryotes, while DNA methylation of promoter sequences has been shown to be a repressive epigenetic mark that down-regulates transcription, DNA methylation is more prevalent within gene bodies and positively correlated with transcript levels (*Jjingo et al., 2012*). Recent studies into *P. falciparum* erythrocytic stages have identified the possible presence of low levels of 5-methylcytosine (5mC) and under-characterized 5-hydroxymethylcytosine (5hmC)-like marks throughout the genome (*Ponts et al., 2013*; *Hammam et al., 2020*; *Nardella et al., 2020*). Presence of low levels of 5mC has also been detected in sporozoites and liver stage forms in *P. vivax* (*Maher et al., 2023*). Due to the GC-poor nature of the *P. falciparum* genome, and the consequential low levels of 5-methylcytosine (5mC) coverage, there remains uncertainty as to what level of transcriptional control is conferred through DNA methylation (*Ponts et al., 2013*). Recent findings suggest perturbation of levels of the DNA methyltransferase PfDNMT2 to have significant impact on transcription and cell proliferation (*Lucky et al., 2023*), but any potential role in antigenic variation and cytoadhesive phenotypes remains to be elucidated.

To deduce the potential gene regulatory relevance of different DNA methylation marks, we first analyzed 5mC and 5hmC levels in NF54CSAh using MeDIP-seq, which allows for differentiation of 5mC from 5hmC due to antibody specificity, compared to prior efforts that have used bisulfite sequencing as the method of choice (*Hammam et al., 2020*). We also investigated the presence of 6-methyladenine (6mA), a common DNA modification in prokaryotes that has only recently been studied in humans and other eukaryotic organisms (*Wion and Casadesús, 2006*; *Zhang et al., 2015*; *Luo et al., 2015*; *Liang et al., 2018*; *Xiao et al., 2018*).

Mapping and peak calling of genome-wide 5mC coverage shows ~94% of a mean 6876 significant consensus peaks mapping to genomic coordinates within the CDS of protein coding genes (*Figure 4A*, *Supplementary file 3*). 5hmC and 6mA coverages were detected at considerably lower levels with only 269 and 0 significant peaks, respectively (*Figure 4—figure supplement 1*, *Supplementary file 3*). We therefore considered that 6mA modifications are most likely absent from the *P. falciparum* genome. 5hmC distribution across the genome does not preferentially map to specific gene families or regions within each gene (*Figure 4—figure supplement 1B*). It is therefore likely that the weak 5hmC signal detected across the genome could be attributed to background noise or 5hmC coverage is non-specific and serves a more generalized function rather than being key to transcriptional regulation.

Due to the more abundant and distinct distribution of 5mC, we then examined the potential correlation between 5mC coverage and transcription. Because no significant difference in exon coverage was detected between NF54CSAh and NF54 in *var2csa* or any other *var* gene (*Figure 4A*), we investigated the difference in distribution of 5mC marks within high versus lowly transcribed genes. We detected a negative correlation for 5mC marks in the promoter of highly transcribed genes, whereas 5mC across

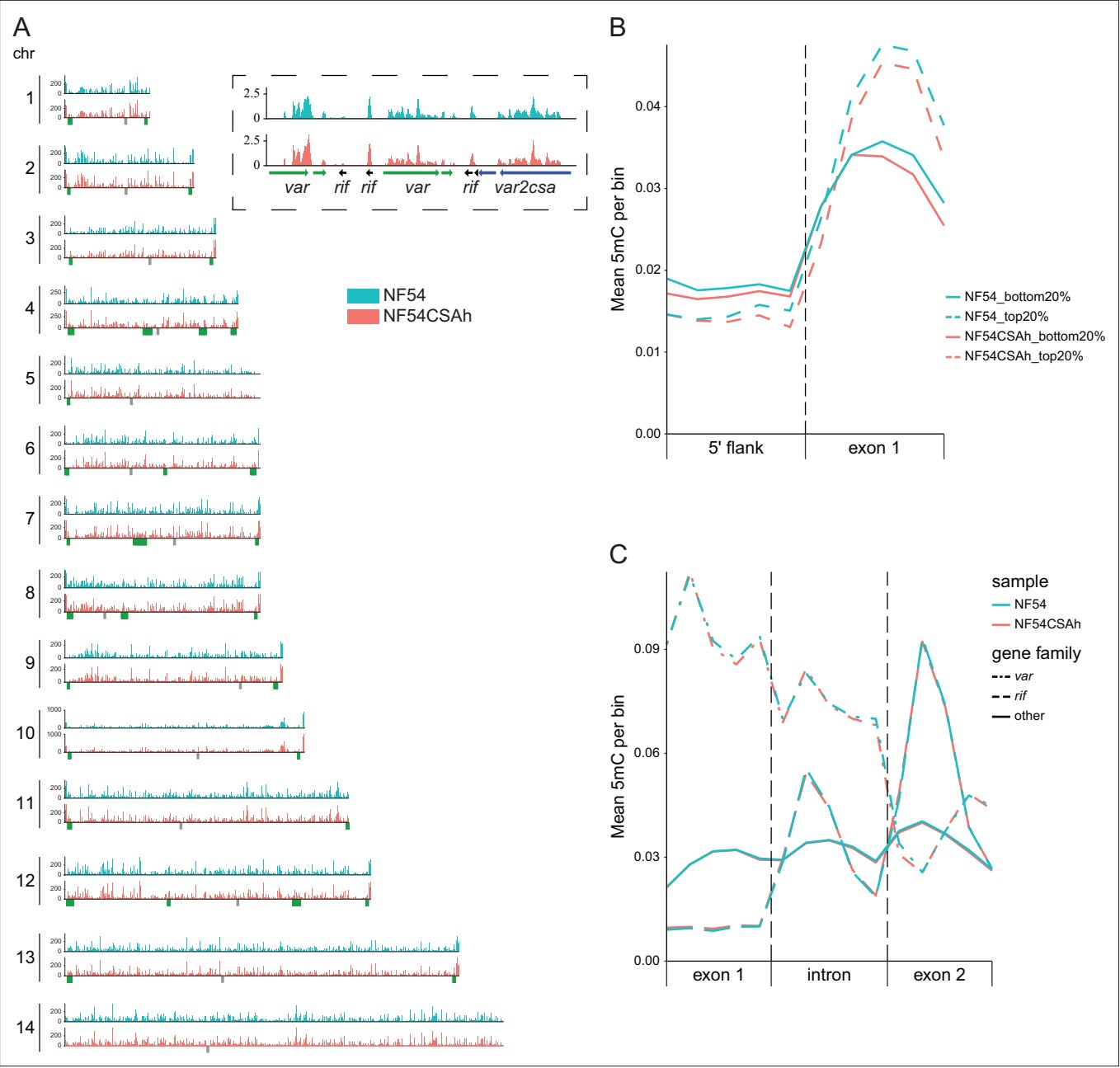

**Figure 4.** Regional differences in 5mC occupancy track with global transcriptional activity but are dissociated from *var2csa* regulation. (**A**) Genome-wide distribution of input normalized MeDIP-seq counts for 5mC in NF54 and NF54CSAh reveal a high level of concordance between the analyzed parasite lines. Regional clusters of antigenically variable genes are denoted in green below each plot with an area of chromosome 12 marked by a dashed box and a zoomed-in insert displaying minor differences in 5mC occupancy in the *var2csa* locus between NF54 and NF54CSAh (*var2csa* in blue, other *var* genes in green and *rif* in black). (**B**) Levels of 5mC marks binned in 5' flanks and first exon for the top 20% highest expressed genes and 20% lowest expressed genes in NF54 and NF54CSAh. (**C**) Distribution of 5mC across exon-intron boundaries for the 5mC enriched *var*, *rif*, and other two-exonic genes.

The online version of this article includes the following figure supplement(s) for figure 4:

**Figure supplement 1.** The *P. falciparum* genome contains very little 5hmC and 6mA compared to 5mC and no pattern between gene type or feature.

the rest of the gene body displays the opposite correlation (*Figure 4B*). Functional differentiation of methylated cytosines within these two distinct regions has already been observed in *P. falciparum* and in other eukaryotic organisms (*Zhang et al., 2006*; *Zemach et al., 2010*; *Ponts et al., 2013*) but not in the context of antigenic variation. Over the years, genome-wide DNA methylation studies have

confirmed this 'DNA methylation paradox' and demonstrated that methylation in exons is found in constitutively expressed genes and has an important role in maintaining transcriptional fidelity by preventing spurious transcription initiation and can affect histone modification and alternative splicing (*Zilberman, 2017*). We therefore investigated the DNA methylation pattern across the exon/intron junction (*Figure 4C*). Altogether, these results confirm the presence of 5mC in the parasite genome within gene bodies of highly expressed genes as well as var genes and their introns. While additional experiments will be needed to further validate the role of DNA methylation in *P. falciparum*, this epigenetic mark does not seem to directly control transcriptional activation and switching of *var2csa* expression but is likely essential in maintaining transcriptional elongation and fidelity, guiding histone modification enzymes and controlling splicing.

## Discussion

The complex and highly variable repertoire of *var* genes constitutes an essential tool for *P. falciparum* pRBCs to express different PfEMP1 proteins in order to cytoadhere to vascular endothelium and avoid the host immune system. The ability to cytoadhere and thereby withdraw from the circulation is crucial for the highly rearranged, rigid, and immunogenic pRBCs to avoid splenic clearance and for the intracellular parasites to survive (*Del Portillo et al., 2012*; *Maier et al., 2008*). Although the prevention of splenic clearance helps the parasite to escape the immune defense of the host, the exposure of the cytoadhesive proteins presents another issue, namely the immune recognition of the adhesins themselves. Therefore, varied expression of these surface proteins through antigenic variation is a necessity, both for the sake of immune escape but also for altered receptor preference and tissue tropism.

Excessive cytoadhesion of *P. falciparum* in the microvasculature of various organs has been strongly associated with malaria disease severity and death. Perhaps the most striking example is the VAR2CSA-mediated sequestration of pRBCs to syncytiotrophoblasts in the placental intervillous space that render previously semi-immune women at risk of severe malaria while pregnant (*Salanti et al., 2004*). The causative *var2csa* gene is unique from other *var* genes in that it is relatively conserved between parasite genomes and is regulated at both a transcriptional and translational level. Parasites cultured continuously in vitro often modify their patterns of *var* gene expression by transcribing *var2csa* without translating the mRNA, as this additional layer of regulation presents the parasites with energy saving and competitive advantages in the absence of the need to cytoadhere for its survival (*Mok et al., 2008*; *Amulic et al., 2009*). Similarly, the complex regulation of the gene allows for the parasites to not express VAR2CSA until infecting a pregnant woman and being presented with the new placental niche.

In the work presented here, we were adamant about analyzing the regulation of *var2csa* in parasites with an appropriate in vivo phenotype instead of parasites with in vitro artifactual and non-productive transcription of the gene. Therefore, several downstream analyses were performed post-phenotypic selection, where we were able to confirm the placental-binding parasites with abundant surface expression of VAR2CSA. In concordance with the adhesive phenotype observed, the selected parasite NF54CSAh showed exclusive and abundant transcription of *var2csa*, which was in stark contrast to the original line which displayed transcript evidence from a vast number of *var* genes. These parasite lines, therefore, presented an excellent opportunity to scrutinize the molecular requirements for *var2csa* selection and activation on a population level, similar to what would occur for isogenic parasites in the in vivo context of PM.

Several studies have previously and conclusively linked gene activation in *P. falciparum* to alterations in DNA accessibility through the change from heterochromatic to euchromatic state (*Lopez-Rubio et al., 2009*; *Bunnik et al., 2018*; *Michel-Todó et al., 2023*; *Lopez-Rubio et al., 2007*; *Salcedo-Amaya et al., 2009*). In line with our transcriptomic findings, which revealed very limited and *var* exclusive differences, we observed a remarkable similarity in global H3K9me3 occupancy between the two analyzed parasite lines. We found H3K9me3 primarily in antigenic variable gene contexts with over-representation in *var*, *rif*, and *stevor*, which highlights the high degree of heterochromatic gene silencing required for these genes. Differential binding analysis revealed only four significant differences between the analyzed lines all centered within or around *var2csa*, with a near absence in H3K9me3 occupancy for NF54CSAh. This clearly confirms the pivotal role of epigenetic regulation through histone modifications in the establishment of the placental-binding phenotype in *P. falciparum*.

Besides histone modifications, other molecular events are known to be involved in the transition of a gene from a heterochromatic to euchromatic state in eukaryotes. One such factor is the macro-molecular events that involve physical repositioning of genes within the nuclear context to allow for their activation or repression (*Ferrai et al., 2010*). In *P. falciparum*, nuclear repositioning has previously been suggested to be involved in regulating the activation of the gametocyte-specific transcription factor locus, pfap2-g, in the transition from the asexual to the early sexual stages in the parasite life cycle progression (*Bunnik et al., 2018*). Nuclear repositioning has also been suggested in antigenic variation (*Ralph et al., 2005*; *Duraisingh et al., 2005*; *Freitas-Junior et al., 2005*; *Coleman et al., 2012*) but its importance has remained somewhat unclear. This is partially due to how well the nuclear context of the parasite is resolved upon interrogation. Previous studies have most often relied on fluorescent in situ hybridizations (FISH) to determine the proximity of genes of interest to a single marker of heterochromatin, such as a subtelomeric repetitive DNA sequence. Considering the small size of the *P. falciparum* nucleus, the naturally close proximity of many *var* genes to these subtelomeric repeats and the limited resolution of fluorescent probe approaches, this is a non-trivial task even in the case of robustly selected parasites with discrete *var* phenotypes. This indeed applies to the subtelo-merically located *var2csa* for which FISH-based endeavors have been inconclusive (*Brolin et al., 2009*; *Ralph et al., 2005*). Our current Hi-C experiments, performed on parasites with vastly different *var* phenotypes, proved able to provide the resolution needed to resolve this matter. We observed a significant decrease in differential interchromosomal interactions for NF54CSAh in the subtelomere of chromosome 12 where *var2csa* is located, whereas interactions around other silent *var* genes were increased. Upon 3D structural modeling, it became evident that the genome of NF54CSAh displays an overall compaction of telomere ends and *var* gene-containing intrachromosomal regions, which is in line with the observed heterochromatic silencing of *var* genes. On the contrary, *var2csa* was distanced from the nearest telomeric cluster to a transcription permissive area of the nucleus. This conclusively suggests that, besides alteration of epigenetic marks, nuclear repositioning and small and localized changes in the chromatin structure are indeed required for activation of *var2csa* transcription. It is important to note that the NF54CSAh line consists of a homogeneous parasite population with respect to var gene expression, whereas the NF54 line is heterogeneous, with parasites expressing distinct var genes. Consequently, interactions surrounding silent var genes and the overall compaction of telomeric ends may be similar in individual parasites within the heterogeneous NF54 population but may appear tighter or more compact in the var2csa-expressing line simply because that population is homogeneous. Lastly, we investigated the differences in DNA methylation between NF54CSAh and NF54. DNA methylation exists in most eukaryotic organisms, including plants, and is involved in various physiological processes and has a complex correlation with gene expression. While the role of promoter hypermethylation in repressing gene transcription has been well documented, emerging evidence indicates that gene body methylation correlates with transcriptional activity in up to 60% of genes in some species. While the presence of low levels of 5mC and 5hmC marks has been identified throughout the *P. falciparum* (*Ponts et al., 2013*; *Hammam et al., 2020*) and *P. vivax* (*Maher et al., 2023*) genomes, the function and the molecular mechanism underlying its regulation have remained obscure. However, recent studies where perturbations of the methylation process by either small molecules (*Nardella et al., 2020*; *Maher et al., 2023*) or reverse genetics (*Lucky et al., 2023*) suggest essentiality and a role in gene regulation. Here, using MeDIP-seq, we show that DNA methylation is mostly absent from the promoter regions of most *var* genes (*Figure 4B*) but is present throughout the CDS of most genes but at a higher level in highly expressed genes (*Figure 4B*). Although there is no clear indication that 5mC regulates the transcriptional activation of *var2csa*, mechanistically, it may be a required modification to guide and stabilize RNA polymerases as well as histone modi-fying enzymes, and ultimately control transcription at the elongation level. We propose that, in *Plasmodium*, DNA methylation may be associated with priming genes for transcriptional activity, rather than repressing transcription. Specifically, higher methylation levels may facilitate recruitment of RNA polymerase II and associated transcriptional machinery to support gene transcription. In *Figure 4B*, we observe higher levels of DNA methylation in the first exon of highly expressed genes in both the NF54 and NF54CSAh lines. Interestingly, we also detect high levels of methylation across most introns of var genes, introns that must be transcribed and are essential for var gene regulation, suggesting a possible sequence-recognition function. While the presence and function of gene body methylation in some species remains controversial, a growing body of evidence suggests that its effects, while

minor, may be shaped by natural selection (*Dixon et al., 2018*) and correlated with fitness (*Muyle et al., 2022*). A high level of DNA methylation in genes known to be involved in antigenic variation in malaria parasites may also be crucial for parasite adaptation and survival in vivo and deserves additional investigation. Collectively, our data confirm a strong association between epigenetics, genome organization, and gene expression in regulating parasite transcripts involved in malaria pathogenicity, but also what appears as a fundamentally different role for DNA methylation through 5mC.

## Materials and methods

### Parasites and in vitro culturing

The *P. falciparum* strains NF54 and NF54CSA-ptefKO (*Chan et al., 2017*) were cultivated according to standard methods (*Trager and Jensen, 1976*). All parasite cultures were maintained in blood group O RBCs at 4% hematocrit in RPMI-1640 medium (Gibco) containing 27 nM NaHCO$_3$ (Sigma), 2 mM L-glutamine (Hyclone), and 2.5 µg/mL gentamicin (Gibco) and supplemented with 10% human A+ serum. The microaerophilic environment was kept constant by gassing of culture flasks with a mixture of 90% N$_2$, 5% O$_2$ and 5% CO$_2$ and cultures were kept in suspension by orbital shaking at 37°C. Parasite cultures were regularly synchronized by treatments with 5% sorbitol as described previously (*Lambros and Vanderberg, 1979*).

### Scanning electron microscopy

To confirm proper pRBC surface morphology prior to the establishment of cytoadhesive phenotype, scanning electron microscopy was used. Magnetically enriched pRBCs were fixed in 2.5% glutaraldehyde (Polysciences), 1% paraformaldehyde (Santa Cruz) in PBS for 1 h, rotating at 4°C. The samples were washed twice with cold PBS. Following the fixation, specimens were adhered onto alcian blue pre-treated 0.45 mm Supor PES membrane filter (Pall Corp.) and washed with 0.1 M phosphate buffer pH7.4 followed by MilliQ water. The membranes were then subjected to stepwise ethanol dehydration, transferred to acetone, and critical-point-dried using carbon dioxide (Leica EM CPD030). The membranes were finally mounted on specimen stubs using carbon adhesive tabs and sputter coated with a 10 nm layer of platinum (Quorum Q150T ES). SEM images were acquired using an Ultra 55 field emission scanning electron microscope (Zeiss) at 3 kV and the SE2 detector.

### Establishment and preservation of parasite phenotypes

The chondroitin sulfate A (CSA) and placental-binding phenotype of NF54CSA was maintained by bi-weekly panning on CSA-coated plastic plates with minor modifications from *Brolin et al., 2009*. Briefly, 100 µg/mL CSA from bovine trachea (Sigma) in phosphate-buffered saline (PBS) was coated on plates overnight leaving a fraction of the plate without CSA-coating. Thereafter, to prevent non-specific binding, plates were blocked with 2% bovine serum albumin (BSA, fraction V, HyClone) in PBS for 1 h. Mature trophozoites were purified using a MACS magnetic cell sorter (Miltenyi BioTec) and resuspended in RPMI-1640 with 10% human serum. Parasites were added to CSA-coated plates and incubated for 1 h at 37°C in microaerophilic conditions with occasional gentle swirling. Plates were thereafter washed with RPMI-1640 until background binding was low, as confirmed by comparison with the non-CSA coated area. Thereafter, the bound pRBCs were recovered from the plates and returned to culture. It is well known that parasites selected to bind to CSA (and other cytoadhesion receptors) revert back to a non-CSA binding state over time. With the selected panning interval, minor reversion was noted. Prior to any downstream analyses, phenotypes were nevertheless confirmed by flow cytometry (see below).

### Flow cytometric analysis of pRBC surface levels of VAR2CSA

For pRBC surface staining of VAR2CSA, cultures with late-stage pRBCs (24–40 hpi) were blocked in PBS with 2% bovine serum albumin (BSA, fraction V, HyClone) for 30 min at room temperature. Primary antibody against the VAR2CSA (goat anti-DBL1-6) was added to blocked cultures at a concentration of 100 µg/mL in PBS with 2% BSA and incubated for 1 h at room temperature. Non-immune goat IgG (Jackson ImmunoResearch) was used for non-specific binding control. Cells were washed twice with 90 µL of PBS before being incubated with a secondary rabbit anti-goat antibody coupled to Alexa647 (Invitrogen) at 1:100 dilution, 5 µg/mL Dihydroethidium (DHE) (Invitrogen) and 10 µg/mL Hoechst

33342 (Invitrogen) in PBS for 1 h at room temperature. Unbound secondary antibody was washed away by 2 × 90 µL washes with PBS before cells were resuspended in PBS to a final hematocrit of 0.3%. 3000 pRBCs per sample were analyzed using a BD FACSVerse (BD Bioscience) and surface positivity was analyzed in FlowJo version 10 by first gating on intact cells on SSC-A and FSC-A, followed by FSC-H and FSC-A for single cells and from there, the Hoechst-DHE double positive pRBC population was selected. This population was analyzed for percentage of Alexa647 positive cells based on a cutoff from the non-immune IgG background binding. The parasite NF54CSA-ptefKO (*Chan et al., 2017*) was used as negative control as the knock-out of the gene encoding the *Plasmodium* Translation Enhancing Factor (PTEF) has previously been shown to be essential for the translation of VAR2CSA. Thus, NF54CSA-ptefKo pRBCs are devoid of VAR2CSA and therefore non-adherent to CSA.

## Placental binding

A placental-binding assay was adapted from *Flick et al., 2001*. Cryo-sections (8 µm in thickness and approximately 6 mm in diameter) from placentas donated by healthy Swedish pregnant women were prepared in a Microm HM 560 cryotome (Thermo Scientific). Sections were mounted onto 10-well microscopy slides (Novakemi AB) and stored at –80°C. Prior to use, the placental sections were submerged in ice-cold PBS. Late-stage pRBCs were enriched using magnetic columns (Miltenyi Biotec) and washed three times before resuspended in binding medium RPMI-1640 (Gibco) with HEPES adjusted to 20 mM, pH 6.8, and 10% human serum at a density of $2.4 \times 10^5$ pRBCs/µL. 25 µL ($0.6 \times 10^7$ cells) of pRBC suspension was added to each placental section and was incubated at 37°C for 1 h in a humid chamber. Unbound cells were washed away by dipping the slides 3 × 5 min in RPMI-1640 with HEPES. Preparations were thereafter fixed with 100% methanol for 15 s and stained with 5% Giemsa (Merck) for 15 min, washed with distilled water, and air-dried. Samples were visualized using an Eclipse 80i (Nikon) microscope at 1000× magnification and blinded for counting. Samples were imaged using an Infinity 3 color camera (Teledyne Lumenera). Number of bound cells per experimental condition was counted for 30–45 fields containing on average 29 pRBCs per field and relative binding was determined compared to the one of VAR2CSA non-expressing pRBCs.

## RNA purification, library preparation, and sequencing

For RNA sequencing, NF54 and NF54CSAh were harvested from three independent selections at 16±2 h post invasion (p.i.) of RBCs. Cell pellets were lysed using five volumes of TRIzol Reagent (Life Technologies) before RNA was purified using the NucleoSpin miRNA Mini kit for miRNA and RNA purification (Macherey-Nagel) without size fractionation, according to the instructions of the manufacturer. The high retention of RNA on the columns of the kit allows for sequential purification without any apparent loss of RNA (not shown). Therefore, all RNA samples were passed over columns in total three times, with the first two elutions being treated with Turbo DNAse (Invitrogen) in solution to ensure a complete lack of contaminating gDNA in the preparations. Strand-specific RNA sequencing (RNA-seq) libraries were generated using the KAPA Stranded RNA-Seq kit (Roche) and were amplified for five PCR cycles (45 s at 98°C for initial denaturing and cycles of 15 s at 98°C, 30 s at 60°C, 30 s at 68°C before final extension for 60 s at 68°C) using the KAPA HiFi HotStart ready mix (KAPA Biosystems) before purification twice over 1X KAPA Pure beads (KAPA Biosystems). Prior to sequencing, the quality of libraries was evaluated using the Agilent 2100 Bioanalyzer (Agilent) and a high sensitivity DNA kit (Agilent). Libraries were quantified using the Collibri Library Quantification kit (Invitrogen) and thereafter sequenced in 75 bp paired-end reads on an Illumina NextSeq550 instrument. PhiX library (Illumina) was used for sequencing control and to diversify libraries.

## RNA-seq data processing and differential expression analysis

Sequenced RNA-seq libraries were assessed for quality using FastQC (v0.11.9). In addition to the index adapters, 12 bp were trimmed from the ends of reads based on sequence quality using Trimmomatic (v0.39) (*Bolger et al., 2014*). Trimmed reads were then aligned to the *P. falciparum* 3D7 genome assembly (PlasmoDB v58) using HISAT2 (v2.1.0) (*Kim et al., 2015*). Samtools (v1.10) (*Li et al., 2009*) was used to filter and sort aligned reads using a quality score of 30 and retrieve properly mapped and paired reads. The output high-quality, properly paired reads were mapped to protein coding genes to retrieve total read count using HTseq-count (v1.99.2) (*Anders et al., 2015*). Differential

gene expression analysis between NF54CSAh and NF54 was performed using DESeq2 (v1.32.0) (*Love et al., 2014*).

## Chromatin immunoprecipitation (ChIP) and ChIP-seq library preparation

For the ChIP-seq experiment, we harvested three biological replicates of NF54 and NF54CSAh (16±2 h p.i) in parallel with the samples collected for RNA-seq. Parasites were freed from RBCs using 0.1% saponin (from quillaja bark, Sigma) and washed thrice with ice-cold PBS. Parasites were cross-linked in 1% methanol-free formaldehyde (Polysciences Inc) at 37°C for 10 min while kept in suspension on an orbital shaker (50 rpm). The cross-linking was thereafter quenched using 125 mM glycine (Sigma) for 5 min at 37°C before cell pellets were washed two times with PBS at 4°C. Nuclei were extracted using a nuclear extraction buffer (10 mM HEPES, 10 mM KCl, 0.1 mM EDTA, 0.1 mM EGTA, 1 mM DTT, 0.5 mM AEBSF, and 1× Halt protease inhibitor cocktail from Thermo Fisher) and incubated for 30 min on ice before addition of Nonidet P 40 substitute (Roche) to a final concentration of 0.5%. Samples were homogenized by passage through 23 G needles three times and 25 G needles seven times. Parasite nuclei were thereafter collected and resuspended in shearing buffer (1% SDS, 10 mM EDTA, 50 mM Tris HCl pH 8.1, and 1x Halt protease inhibitor cocktail, Thermo Fisher). Chromatin was fragmented using the Covaris ultra sonicator (ME220) for 5 min with the following settings: 25% duty cycle, 75 W intensity peak incident power, 1000 cycles per burst. Insoluble material was removed by centrifugation for 10 min at 15700 rcf at 4°C before samples were diluted 10-fold in ChIP dilution buffer (16.7 mM Tris–HCl pH 8, 1.2 mM EDTA, 0.01% SDS, 150 mM NaCl, 1.1% Triton, and 1× Halt protease inhibitor cocktail from Thermo Fisher). To reduce non-specific background, samples were precleared with ChIP-grade protein A/G magnetic beads (Thermo Scientific)/0.2 mg/mL Salmon Sperm DNA (Invitrogen) for 2 h at 4°C with end-to-end rotation. To retrieve H3K9 trimethylated histones, samples were incubated with 2 µg anti-H3K9me3 antibodies (ab8898, Abcam) or a non-immune rabbit IgG as negative control (12-370, Upstate) overnight at 4°C. An input sample was saved at 4°C until the de-crosslinking step. Antibody–protein complexes were recovered during 2 h incubation at 4°C with protein A/G magnetic beads (Thermo Scientific), followed by duplicate washes of 15 min with low-salt wash buffer (0.1% SDS, 1% Triton X-100, 2 mM EDTA, 20 mM Tris–HCl pH 8.1, 150 mM NaCl), high-salt wash buffer (0.1% SDS, 1% Triton X-100, 2 mM EDTA, 20 mM Tris–HCl pH 8.1, 500 mM NaCl), LiCl wash buffer (0.25 M LiCl, 1% NP-40, 1% sodium deoxycholate, 1 mM EDTA, 10 mM Tris–HCl pH 8.1), and TE buffer (10 mM Tris–HCl pH 8, 1 mM EDTA). Chromatin immuno complexes were thereafter eluted from the beads twice with elution buffer (1% SDS, 0.1 M NaHCO$_3$) for 15 min at room temperature, and elutes were combined. RNA was removed by incubation with 0.48 µg/µL RNase A (Invitrogen) for 30 min at 37°C, followed by a 2 h incubation at 45°C with proteinase K (final concentration 0.24 µg/µL, Ambion). Samples were de-crosslinked overnight at 65°C by adding NaCl to a final concentration of 200 mM. DNA was thereafter extracted using phenol:chloroform:isoamyl alcohol (25:24:1, Sigma) and ethanol precipitation and was further purified with KAPA Pure beads (KAPA Biosystems).

ChIP-seq libraries were prepared using the KAPA LTP library preparation kit (KAPA Biosystems). Immunoprecipitated and input libraries were amplified for 16 and 5 PCR cycles, respectively (45 s at 98°C for initial denaturing and cycles of 15 s at 98°C, 30 s at 60°C, 30 s at 68°C before final extension for 60 s at 68°C) using the KAPA HiFi HotStart ready mix (KAPA Biosystems) before purification twice over 1X KAPA Pure beads (KAPA Biosystems). Prior to sequencing, the quality of libraries was evaluated using the Agilent 2100 Bioanalyzer (Agilent) and a High Sensitivity DNA kit (Agilent). Libraries were quantified using the Colibri Library Quantification kit (Invitrogen) and thereafter sequenced in 75 bp paired-end reads on an Illumina NextSeq550 instrument. PhiX library (Illumina) was used for sequencing control and to diversify libraries.

## Methylated DNA immunoprecipitation (MeDIP) and MeDIP-seq library preparation

Biological triplicates of NF54 and NF54CSAh (16±2 h p.i) were harvested for MeDIP-seq in parallel with the samples collected for RNA-seq and ChIP-seq. Parasites were initially released from host RBCs by 0.1% saponin (from quillaja bark, Sigma) and washed thrice with ice-cold PBS. Thereafter, genomic DNA was extracted using QIAamp DNA Blood Kit (QIAGEN) following the instructions of the manufacturer and including the optional RNase A treatment. Extracted DNA was eluted in NE buffer (Macherey-Nagel) and quantified with dsDNA HS assay kit (Invitrogen) on a Qubit 3 fluorometer

(Invitrogen). gDNA was fragmented in TE buffer using the Covaris ultra sonicator (ME220) for 130 s with the following settings: 20% duty cycle, 70 W intensity peak incident power, 1000 cycles per burst, resulting in average fragment size of approximately 300 bp. Thereafter, gDNA was precipitated and concentrated by addition of 0.1 volumes of sodium acetate (Alfa Aesar) and 3 volumes of 100% ethanol. gDNA was quantified with dsDNA HS assay kit (Invitrogen) for Qubit 3 fluorometer (Invitrogen) and 100/30 ng (immunoprecipitation/input) used for library preparation using KAPA LTP library preparation kit (KAPA Biosystems). Library preparation was stopped prior to PCR amplification. The quality of libraries was assessed with Agilent 2100 Bioanalyzer (Agilent) and High Sensitivity DNA kit (Agilent) and quantity with the Collibri Library Quantification kit (Invitrogen). Thereafter, equal amounts of libraries from NF54 and NF54CSAh were pooled for the same bio-replicate for the same antibody. Libraries were denatured at 95°C for 10 min and then incubated on ice for 10 min. The Methylated DNA Immunoprecipitation (MeDIP) Kit (Active Motif) was used according to the instructions by the manufacturer with some modifications. 3 µg of polyclonal rabbit anti-5mC (#61255, Active Motif) and rabbit IgG (#103524, Active Motif) were added to the denatured libraries. For the elucidation of potential presence of 5hmC and 6mA in NF54CSAh only, 3 µg of polyclonal rabbit anti-5mhC (#39069, Active Motif) and polyclonal rabbit anti-6mA (#202003, Synaptic Systems) was used respectively. Immune complexes were collected using 20 µL of protein G magnetic beads (Active Motif). After elution from the beads, samples were purified using 1X KAPA Pure beads (KAPA Biosystems) and PCR amplified using the KAPA HiFi HotStart ready mix (KAPA Biosystems) for eight PCR cycles (45 s 98°C of initial denaturing and amplification at 15 s at 98°C, 30 s at 60°C, 30 s at 68°C and final extension 60 s at 68°C). Libraries were analyzed for quality using an Agilent 2100 Bioanalyzer (Agilent) and the High sensitivity DNA kit (Agilent), quantified using Collibri Library Quantification kit (Invitrogen) and thereafter sequenced in 75 bp paired-end reads on an Illumina NextSeq550 instrument. PhiX library (Illumina) was used for sequencing control and to diversify libraries.

## ChIP-seq and MeDIP-seq data processing and peak calling

ChIP-seq reads in fastq format were assessed for quality using FastQC (v0.11.9). Index adapters and an additional 8 bp were trimmed from the ends of reads based on sequence quality using Trimmomatic (v0.39) (*Bolger et al., 2014*). Bowtie2 (v2.3.5.1) (*Langmead and Salzberg, 2012*) was used to align reads to the *P. falciparum* 3D7 genome (PlasmoDB v58). PCR duplicates were tagged using Picardtools (v2.26.11) (http://broadinstitute.github.io/picard). Quality filtering using a mapping score of 30, sorting, and indexing were performed using Samtools (v1.10) (*Li et al., 2009*) keeping only properly mapped and paired reads. Output properly paired, high-quality, deduplicated reads were then mapped to the genome using Bedtools (v2.27.1) (*Quinlan and Hall, 2010*) to retrieve per base coverage genome wide. All three sets of reads for each sample (H3K9me3, IgG, and input) were then normalized by read count at each locus per million mapped reads before subtracting IgG and input read counts from H3K9me3 to remove background noise for visualizing chromosomal coverage. Genome-wide coverage was binned at 10 bp resolution prior to mapping. Significant peaks were called using MACS3 (v3.0.0a7) (https://github.com/macs3-project/MACS; *Liu et al., 2023*) with FDR <0.05 for calling composite broad peaks. TSS coverage profiles were mapped using deeptools2 (v3.5.1) (*Ramírez et al., 2016*) bamCompare to subtract input reads and regions from 500 bp 5' of the TSS to 1500 bp 3' of the TSS were scored with a 1 bp bin size. DiffBind (v3.2.7) was utilized for differential peak calling between NF54CSAh and NF54.

MeDIP-seq data processing was performed the same as above, using the same versions of FastQC, Trimmomatic, Picardtools, Samtools, Bedtools, and MACS3. Counts per million normalization was performed prior to subtracting IgG and input reads from 5mC, 5hmC, and 6mA for genome-wide visualization at 10 bp resolution. MACS3 peak calling was performed using the default narrow peak calling for 5mC, 5hmC, and 6mA, but minimum fold coverage for model building was reduced to 2. Default DiffBind parameters were used for differential peak calling between NF54CSAh and NF54 for 5mC, and between NF54CSAh 5mC, 5hmC, and 6mA.

## Chromosome conformation capture sequencing (Hi-C) library preparation

Three biological replicates of NF54 and NF54CSAh (at 20±2 h.pi.) were fixed with 1.25% methanol-free formaldehyde (Polysciences Inc) for 25 min at 37°C and quenched with 150 mM glycine for 15 min

at 37°C. Thereafter, samples were incubated at 4°C on a rocking platform for 15 min. Parasites were collected and washed with ice-cold PBS. The supernatant was removed and parasite pellets stored at –80°C. Crosslinked parasite pellets were resuspended in lysis buffer (10 mM Tris–HCl, pH 8.0, 10 mM NaCl, 2 mM AEBSF, 0.10% Igepal CA-360 [v/v], and 1X protease inhibitor cocktail) and incubated for 30 min on ice. Samples were homogenized by a 26.5-gauge needle, then washed in lysis buffer. Pellets were resuspended in 0.5% SDS and incubated at 62°C to solubilize the chromatin. DNA was digested overnight with 100 units MboI (NEB) restriction enzyme, then the ends filled using dTTP, dGTP, dATP, biotinylated dCTP, and 25 units DNA polymerase I. Blunt ends were then ligated using 4000 units T4 DNA ligase and chromatin de-crosslinked using decrosslinking buffer (50 mM Tris–HCl, pH 8.0, 1% SDS, and 500 mM NaCl) and RNase A. Ligated DNA was purified and sheared to a length of ~300–500 bp using the Covaris S220 sonicator (settings: 10% duty factor, 140 peak incident power, and 200 cycles per burst for 65 s). Fragments were pulled down using streptavidin T1 beads (Invitrogen) then library prepped by end-repair, A-tailing, and adapter ligation all in lo-bind tubes. Libraries were PCR amplified with NEB multiplex oligos (45 s at 98°C, 12 cycles of 15 s at 98°C, 30 s at 55°C, 30 s at 62°C and a final extension step of 5 min at 62°C) and sequenced using the NOVASeq platform (Illumina).

### Hi-C data processing, differential interaction analysis and 3D modeling

Paired-end Hi-C libraries were processed (mapping, read pairing, quality filtering, binning, and normalizing) using HiC-Pro (v3.1.0) (*Servant et al., 2015*) with minimum mapping quality of 30 at 10 kb resolution when aligning to the *P. falciparum* 3D7 genome (PlasmoDB v58). Interaction heatmaps were generated using the ICED-normalized interaction matrices after an additional per-million read count normalization to allow for direct comparison between NF54CSAh and NF54. Correlation among biological replicates was evaluated with HiCRep (*Yang et al., 2017*), then replicates were merged to generate a single representative sample. To enhance visualization, all intra-bin contacts and contacts within a two-bin distance were set to the 90th percentile, reducing the range of the data needing to be color mapped. Regions of differential interactions between NF54CSAh and NF54 were identified using Selfish (*Ardakany et al., 2019*) for all intrachromosomal and interchromosomal matrices. Coordinate matrices were generated by PASTIS (*Varoquaux et al., 2014*) from raw read count matrices and then visualized as 3D chromatin models in ChimeraX (*Goddard et al., 2018*) while highlighting bins containing *var* genes, telomeres, and centromeres.

## Acknowledgements

The authors would like to thank Lars Haag at the Electron Microscopy Unit (Emil), Karolinska Institutet, for excellent technical assistance relating to the scanning electron microscopy. This work was funded by The Knut and Alice Wallenberg Foundation (KAW 2017.0055 to UR); The Swedish Research Council (VR 2018-05814, VR 2016-02917 to UR); The National Institutes of Allergy and Infectious Diseases and the National Institutes of Health (R01 AI136511, R21 AI142506 to KLR); and The University of California, Riverside (NIFA-Hatch-225935 to KLR). Funding for open access charge: The Swedish Research Council.

## Additional information

### Funding

| Funder | Grant reference number | Author |
| --- | --- | --- |
| National Institutes of Health | R01 AI136511 | Karine G Le Roch |
| National Institutes of Health | R21 AI142506 | Karine G Le Roch |
| Knut and Alice Wallenberg Foundation | KAW 2017.0055 | Ulf Ribacke |
| Swedish Research Council | VR 2018-05814 | Ulf Ribacke |

| Funder | Grant reference number | Author |
| --- | --- | --- |
| Swedish Research Council | VR 2016-02917 | Ulf Ribacke |
| University of California, Riverside | NIFA-Hatch-225935 | Karine G Le Roch |

The funders had no role in study design, data collection and interpretation, or the decision to submit the work for publication.

## Author contributions

Todd Lenz, Formal analysis, Visualization, Methodology, Writing – original draft, Writing – review and editing; Madle Sirel, Investigation, Methodology; Hannes Hoppe, Sulman Shafeeq, Investigation; Karine G Le Roch, Conceptualization, Resources, Data curation, Formal analysis, Supervision, Funding acquisition, Validation, Writing – original draft, Project administration, Writing – review and editing; Ulf Ribacke, Conceptualization, Data curation, Formal analysis, Supervision, Funding acquisition, Investigation, Methodology, Writing – original draft, Project administration, Writing – review and editing

### Author ORCIDs

Sulman Shafeeq ⓘ https://orcid.org/0000-0003-1152-526X
Karine G Le Roch ⓘ https://orcid.org/0000-0002-4862-9292

Reviewer #2 (Public review): https://doi.org/10.7554/eLife.93632.3.sa1
Reviewer #3 (Public review): https://doi.org/10.7554/eLife.93632.3.sa2
Author response https://doi.org/10.7554/eLife.93632.3.sa3

# Additional files

## Supplementary files

Supplementary file 1. RNA-seq analysis.

Supplementary file 2. ChiP-seq analysis.

Supplementary file 3. MeDIP-seq analysis.

MDAR checklist

## Data availability

No new algorithms or tools were created to process or visualize data contained in this manuscript and any scripts used in the analysis of data use previously established packages. Scripts used to process data and generate figures for this manuscript are freely available on the personal GitHub page of Todd Lenz (https://github.com/tlenz88/). Data processing and plotting of RNA-seq, ChIP-seq, and MeDIP-seq data was accomplished using the SimpleSeq pipeline with accompanying tools EasyDGE and EasyPeaks found on the SimpleSeq repository (*Lenz, 2026*). Plotting of HiC heatmaps and 3D models was accomplished using the scripts found on the HiCplotter (*Lenz, 2025a*) and Chrom3D (*Lenz, 2025b*) repositories, respectively. The sequencing datasets generated and analyzed during the current study are available in the SRA repository under the accession number PRJNA947327.

The following dataset was generated:

| Author(s) | Year | Dataset title | Dataset URL | Database and Identifier |
| --- | --- | --- | --- | --- |
| Le Roch Lab | 2023 | Epigenetics and chromatin structure regulate var2csa expression and the placental binding phenotype in *Plasmodium falciparum* | https://www.ncbi.nlm.nih.gov/bioproject/PRJNA947327 | NCBI BioProject, PRJNA947327 |

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
