## [Editor Report · eLife Assessment]

This interesting study presents a multi-OMICs approach to unify different lines of evidence regarding the epigenetic regulation of the key virulence factor causing placental malaria during *P. falciparum* infection. Most results are confirmatory of previous observations; nonetheless, the claims are supported by **convincing** evidence. The combinatorial approach chosen here is unprecedented and therefore provides **valuable** new data. In addition, the comparative investigation of different DNA methylation modifications is novel and disproves a direct role in *var* gene regulation.

---

## [Referee Report · Reviewer #2 (Public review)]

Summary:

Dr Lenz and colleagues report on their in vitro studies comparing gene transcription and epigenetic modifications in *Plasmodium falciparum* NF54 parasites selected or not selected for adhesion of the infected erythrocytes (IEs) to the placental IE adhesion receptor chondroitin sulfate A (CSA).

The authors report that selection led to preferential transcription of var2csa, the gene that encodes the VAR2CSA-type PfEMP1 well-established as the PfEMP1 mediating IE adhesion to CSA. They confirm that transcriptional activation of var2csa is associated with distinct depletion of H3K9me3 marks and that transcriptional activation is linked to repositioning of var2csa. Finally, they provide preliminary evidence potentially implicating 5mC in transcriptional regulation of var2csa.

Strengths:

The study confirms previously reported features of gene transcription and epigenetic modifications in *Plasmodium falciparum*.

Weaknesses:

No major new finding is reported.

Comments on revisions:

I suggest replacing the term "pregnancy-associated malaria (PAM)" with the more current and more precise term "placental malaria (PM)" throughout the manuscript.

L. 59-60: "... shielding of the parasite antigens expressed on pRBC surfaces by leukocytes...". It is unclear to me what this means - I suggest a rephrasing for improved clarity.

L. 144-6: Please provide a reference for the primary antibody reagent used.

---

## [Referee Report · Reviewer #3 (Public review)]

Summary:

The manuscript by Lenz et al. seeks to investigate molecular mechanisms directing virulence gene expression in the malaria parasite *Plasmodium falciparum*. The report provides a detailed characterization of the phenotypic and epigenetic features of a var2csa expressing parasite population, the key virulence gene causing the clinical syndrome of placental malaria. Novel evidence supporting the concept that active expression of this gene is associated with nuclear repositioning away from suppressive regions of chromatin is presented. In addition, the authors conducted a preliminary characterization of different forms of DNA methylation, suggesting that 5-methylcytosine is enriched in virulence genes, but does not correlate with their activation or repression. However, a trend towards higher enrichment of 5-methylcytosine in highly active as opposed to inactive genes from the core genome was reported, although this observation requires further validation.

Strengths:

The concise study provides a well documented and controlled set of experiments utilizing state-of-the-art OMICs methodologies including ChIPseq, RNAseq, chromatin-conformation capture (Hi-C) and DNA methylation (MeDIPseq) to generate deep insight into the epigenetic regulation of the key virulence factor of *P. falciparum*. The study unifies different lines of evidence and thereby contributes to a clearer understanding of the mechanisms underlying active expression of var2csa.

Weaknesses:

Although all experiments appear to have been rigorously conducted and documented with appropriate replicates and controls, the study is overall lacking statistical support from individual analyses of the biological replicates. In particular, the key novel result suggesting increased distance of the active var2csa gene from regions of heterochromatin as assessed by chromatin conformation capture would benefit from further analysis by comparison with other genetic loci. This also applies to the differential DNA methylation patterns, which should be dissected in more detail to support any association with gene expression or intron function.

---

## [Author Response]

The following is the authors’ response to the original reviews

**Public Reviews:**

**Reviewer #1 (Public Review):**
Summary:The manuscript by Lenz and colleagues describes a detailed examination of the epigenetic changes and alterations in subnuclear arrangement associated with the activation of a unique var gene associated with placental malaria in the human malaria parasite *Plasmodium falciparum*. The var gene family has been heavily studied over the last couple of decades due to its importance in the pathogenesis of malaria, its role in immune avoidance, and the unique transcriptional regulation that it displays. Aspects of how mutually exclusive expression is regulated have been described by several groups and are now known to include histone modifications, subnuclear chromosomal arrangement, and in the case of var2csa, regulation at the level of translation. Here the authors apply several methods to confirm previous observations and to consider a possible role for DNA methylation. They demonstrate that the histone mark H3K9me3 is found at the promoters of silent genes, var2csa moves away from other var gene clusters when activated, and while DNA methylation is detectable at var genes, it does not seem to correlate with transcriptional activation/silencing. Overall, the data and approach appear sound.Strengths:The authors employ the latest methods for epigenetic analysis of histone marks, transcriptomic analysis, DNA methylation, and chromosome conformation. They also use strong selection pressure to be able to examine the gene var2csa in its active and silent state. This is likely the only paper that has used all these methods in parallel to examine var gene regulation. Thus, the paper provides readers with confidence in the interpretation of independent methods that address a similar subject.

We thank the reviewer for this positive assessment. We appreciate the recognition that our study combines complementary approaches including histone mark profiling, transcriptomic analysis, DNA methylation mapping, and chromosome conformation capture in parallel to the use of strong population selection that enables a controlled comparison of var2csa in active versus silent states. We agree that the convergence of independent methods strengthens confidence in the interpretation.

Weaknesses:The primary weakness of the paper is that none of the conclusions are novel and the overall conclusions do not shed much new light on the topic of var gene regulation or antigenic variation in malaria parasites. The paper is largely confirmatory. The roles of H3K9me3 and subnuclear localization in var gene regulation are well established by many groups (including for var2csa), albeit in some cases using alternative methods. The only truly unique aspect of the manuscript is the description of 5mC at var2csa when the gene is transcriptionally active or silent. Here the authors demonstrate that the mark has no clear role in transcriptional activation or silencing, however, this will not be surprising to many in the field who have previously cast doubt on a regulatory role for this modification.

While we agree that some individual features of var gene regulation, including H3K9me3 enrichment, have been described previously, our study integrate for the first time several layer of gene regulation on the clinically important var2csa locus using phenotypically homogeneous placental-binding parasite populations. As expected, var2csa activation coincided with a loss of H3K9me3 at the locus. However, using high-resolution chromatin conformation capture (to our knowledge, this experiment had never been applied to phenotypically homogeneous parasite populations), we quantified the repositioning of var2csa relative to heterochromatic telomeric clusters. We further assessed DNA methylation in this framework and show that 5-methylcytosine is broadly present at var genes and may correlate with transcript level, but is uncoupled from transcriptional activation, repression, and switching. Together, these findings integrate transcriptional state, chromatin marks, and 3D genome organization at var2csa and argue against models in which 5mC acts as a primary regulatory switch for var gene expression.

**Reviewer #2 (Public Review):**
Summary:Dr Lenz and colleagues report on their in vitro studies comparing gene transcription and epigenetic modifications in *Plasmodium falciparum* NF54 parasites selected or not selected for adhesion of the infected erythrocytes (IEs) to the placental IE adhesion receptor chondroitin sulfate A (CSA).The authors report that selection led to preferential transcription of var2csa, the gene that encodes the VAR2CSA-type PfEMP1 well-established as the PfEMP1 mediating IE adhesion to CSA. They confirm that transcriptional activation of var2csa is associated with distinct depletion of H3K9me3 marks and that transcriptional activation is linked to repositioning of var2csa. Finally, they provide preliminary evidence potentially implicating 5mC in the transcriptional regulation of var2csa.Strengths:The study confirms previously reported features of gene transcription and epigenetic modifications in *Plasmodium falciparum*.

As stated in our response to Reviewer 1, our study combines, for the first time, complementary approaches, including transcriptomic analysis, histone mark profiling, DNA methylation mapping, and chromosome conformation capture, together with strong population selection to enable a controlled comparison of var2csa in active versus silent states.

Weaknesses:No major new finding is reported. The strength of the evidence presented is mostly solid, although certain elements, e.g., the role of 5mC in transcriptional regulation of var2cs, appear preliminary and incomplete.

While we agree that no major new finding is reported, we were able to use for the first time a high-resolution chromatin conformation capture method to quantify the repositioning of var2csa relative to heterochromatic telomeric clusters. We also further assessed that 5-methylcytosine is present at var genes and may correlate with transcript level, but is uncoupled from transcriptional activation, repression, and switching. Together, these findings integrate for the first time transcriptional state, chromatin marks, and 3D genome organization at var2csa and argue against models in which 5mC acts as a primary regulatory switch for var gene expression.

**Recommendations for the authors:**

**Reviewer #1 (Recommendations for the Authors):**
(1) In the second paragraph of the introduction, the authors state "....such as the shielding of the parasite antigens expressed on pRBC surfaces by other cells and the evasion of splenic clearance (8)." What does "other cells" mean here?

We thank the reviewer for this comment. We have clarified the cell type in the text.

(2) In their interpretation of the Hi-C data, the authors conclude that the var2csa expressing parasites display "tighter heterochromatin control of var gene regions" and "interactions around other silent var genes were increased" and "an overall compaction of telomere ends and var gene-containing intrachromosomal regions". While the data appear to show that this is true when they compare the two parasite populations, I am concerned that the authors might be misinterpreting the data. It is important to note that the NF54CSAh line is heavily selected to be nearly entirely homogeneous for var gene expression while the NF54 line is exceptionally heterogeneous. This is shown in Figure 1G. Thus, any chromosomal arrangement specific for var gene expression in the unselected NF54 population will be similarly heterogeneous and therefore could appear less tight. In other words, interactions around silent var genes and overall compaction of telomere ends might be identical between individual parasites within these populations, but appear tighter or more compact in the var2csa expressing line simply because it is a homogeneous population. Perhaps this is what the authors meant to convey, however as currently written, it seems that they conclude the expression of var2csa results in a unique change in chromosome organization. A better comparison would be two populations homogeneously expressing different var genes, one expressing var2csa and one expressing an alternative var gene. Such lines can be generated through clonal isolation or selection for binding to a different host receptor.

We thank the reviewer for this comment. The reviewer is correct, and we have revised the Discussion section of the manuscript to clarify this issue.

(3) The title of the last section of the Results is "Distribution of DNA methylation influences gene expression overall but does not mediate transcriptional activation and switching in antigenic variation". This is an overstatement. The authors show that DNA methylation is absent at var gene promoter regions and enriched in coding regions, but there they provide no evidence that it "influences gene expression overall". This is speculation. Lastly, when the authors examined 5mC occupancy across genes, did they normalize for GC content of the DNA sequences? GC content is known to increase dramatically in coding regions (particularly in var genes) and thus could explain the distribution of this mark. If the authors corrected for this, they should directly state this in the results section. If they did not, they should explain why they don't think this property of the *P. falciparum* genome explains the distribution of 5mC.

There is often a misconception in the field that DNA methylation is primarily confined to CpG islands in promoter regions and functions mainly as a repressor of transcription. However, in contrast to promoter methylation, methylation within gene bodies is generally associated with higher levels of gene expression, suggesting a role in facilitating transcription elongation. Gene-body methylation can also repress internal promoters, thereby preventing spurious transcription initiation within the gene. In addition, it has been shown to influence alternative splicing by affecting RNA polymerase II elongation kinetics.

We propose that, in *Plasmodium*, DNA methylation may be associated with priming genes for transcriptional activity rather than repressing transcription. Specifically, higher methylation levels may facilitate recruitment of the RNA polymerase II transcriptional machinery to enable transcription. In Figure 4B, we observe higher levels of DNA methylation in the first exon of highly expressed genes in both the NF54 and NF54CSAh lines. Interestingly, we also detect high levels of methylation across most introns of the *var* genes, introns that must be transcribed, cannot be degraded, and are essential for *var* gene regulation, suggesting a possible sequence-recognition function. We have edited the manuscript to improve clarity.

(4) In the legend to Figure 3D, the authors state that the centromeres are shown in blue, however in the figure they appear to be grey while var2csa is blue.

We have revised the figure legend accordingly.

**Reviewer #2 (Recommendations For The Authors):**
I recommend using the term "transcription" rather than "expression" when discussing events at the gene level.

We have revised the manuscript accordingly.

I also recommend using the term "adhesion" to describe the physical interaction between infected erythrocytes and adhesion receptors rather than adherence", which should be reserved to describe non-physical affinity (e.g., beliefs, faith).

We have revised the manuscript accordingly.

Important new evidence regarding transcriptional regulation of var genes in general and var2csa in particular should be discussed and cited.

We have revised the manuscript accordingly.